# RNA Sequencing in COVID-19 patients identifies neutrophil activation biomarkers as a promising diagnostic platform for infections

Richard Wargodsky[1☯], Philip Dela Cruz[2☯], John LaFleur[3], David Yamane[2,3], Justin Sungmin Kim[2], Ivy Benjenk[2], Eric Heinz[2], Obinna Ome Irondi[2], Katherine Farrar[2], Ian Toma[1,4,5], Tristan Jordan[1], Jennifer Goldman[1], Timothy A. McCaffrey[1,4,5,6]*

1 Department of Medicine, Division of Genomic Medicine, The George Washington University Medical Center, Washington, DC, United States of America, 2 Department Anesthesiology and Critical Care Medicine, The George Washington University Medical Center, Washington, DC, United States of America, 3 Department of Emergency Medicine, The George Washington University Medical Center, Washington, DC, United States of America, 4 Department of Clinical Research and Leadership The George Washington University Medical Center, Washington, DC, United States of America, 5 True Bearing Diagnostics, Washington, DC, United States of America, 6 Department of Microbiology, Immunology, and Tropical Medicine, The George Washington University Medical Center, Washington, DC, United States of America

☯ These authors contributed equally to this work.
* mcc@gwu.edu

**Data Availability Statement:** The expression-level data (raw RPKM) is deposited in the Gene Expression Omnibus (GEO) at the accession

## Abstract

Infection with the SARS-CoV2 virus can vary from asymptomatic, or flu-like with moderate disease, up to critically severe. Severe disease, termed COVID-19, involves acute respiratory deterioration that is frequently fatal. To understand the highly variable presentation, and identify biomarkers for disease severity, blood RNA from COVID-19 patient in an intensive care unit was analyzed by whole transcriptome RNA sequencing. Both SARS-CoV2 infection and the severity of COVID-19 syndrome were associated with up to 25-fold increased expression of neutrophil-related transcripts, such as neutrophil defensin 1 (DEFA1), and 3-5-fold reductions in T cell related transcripts such as the T cell receptor (TCR). The DEFA1 RNA level detected SARS-CoV2 viremia with 95.5% sensitivity, when viremia was measured by ddPCR of whole blood RNA. Purified CD15+ neutrophils from COVID-19 patients were increased in abundance and showed striking increases in nuclear DNA staining by DAPI. Concurrently, they showed >10-fold higher elastase activity than normal controls, and correcting for their increased abundance, still showed 5-fold higher elastase activity per cell. Despite higher CD15+ neutrophil elastase activity, elastase activity was extremely low in plasma from the same patients. Collectively, the data supports the model that increased neutrophil and decreased T cell activity is associated with increased COVID-19 severity, and suggests that blood DEFA1 RNA levels and neutrophil elastase activity, both involved in neutrophil extracellular traps (NETs), may be informative biomarkers of host immune activity after viral infection.

#GSE189990. To protect the anonymity of the research subjects, the raw sequence files from this study will be provided to qualified investigators that can ensure compliance with appropriate IRB and HIPAA regulations for any future data usage. At the request of the Investigators, the GWU IRB, and the GW Office of Research Integrity, access to sequence-level data will be made available by contacting the GWU IRB at orhirb@email.gwu.edu. The human genome files for alignment were obtained from UCSC at this link for HG38 (https://hgdownload.soe.ucsc.edu/goldenPath/hg38/bigZips/).

**Funding:** The authors are grateful for the generous financial support of True Bearing Diagnostics, Inc., The St. Laurent Institute, and The Ulvi and Reykhan Kasimov Family. The authors are also grateful for the institutional support provided by the CTSI-CN Award Number UL1TR001876 from the NIH National Center for Advancing Translational Sciences, and the Core Instrument Grant for the Bio-Rad ddPCR S10 OD021622. The funders had no role in study design, data collection and analysis, decision to publish, or preparation of the manuscript.

**Competing interests:** TM and IT have an equity interest in True Bearing Diagnostics, Inc., a diagnostics company developing RNA biomarkers for various diseases, including coronary artery disease and internal infections. TM is seeking patent protection for technology related to the current studies. This does not alter our adherence to PLOS ONE policies on sharing data and materials. The other authors declare there are no competing interests.

**Abbreviations:** CBC, complete blood count; ddPCR, droplet digital PCR; DEGs, differentially expressed genes; RIN, RNA integrity number; RNAseq, RNA sequencing; rRNA, ribosomal RNA; RPKM, reads per kilobase of exon per million mapped total reads.

## Introduction

The COVID-19 pandemic, initiated by SARS-CoV2 viral infection, illustrates the need for the rapid detection of novel and variant pathogens, as well as specific risk prediction models to identify patients at high risk of severe or fatal disease. Unfortunately, clinical risk models for critically ill patients have been only modestly accurate in predicting outcomes for COVID-19 patients [1]. Furthermore, similar to septic shock, there is an incomplete understanding of the relative contributions of the pathogen load versus a hyperactive immune response by the host. The detection of SARS-CoV2 viral load has mostly been measured only in the accessible regions of the respiratory tract, and requires prior knowledge of the viral sequence to achieve relatively specific PCR amplification. The current studies investigated a complementary strategy that utilizes the host immune system to report a pathogen's presence and activity. Truly 'agnostic' blood biomarkers of infection would have numerous uses, especially in helping to discriminate infectious diseases from aseptic inflammation, such as might occur in asthma, allergies, fibrotic, or autoimmune disorders.

In previous work, we employed whole transcriptome profiling to identify RNA biomarkers in peripheral blood from patients with appendicitis or respiratory infections [2]. Relatively large fold-change increases (~20-fold) were identified in RNAs coding for neutrophil primary granule markers, such as neutrophil defensins (DEFA1), in infections where the pathogen was directly accessible to the immune system, such as pneumonia. In appendicitis, where the infection can take the form of a 'biofilm' that is not directly accessible to the immune cells [3], there was marked increases in mRNAs coding for IL8 receptor-ß (IL8RB/CXCR2), and secondary granule proteins such as alkaline phosphatase (ALPL) [2]. Thus, it was hypothesized that the RNA biomarkers would report immune activation in response to a SARS-CoV2 infection, regardless of the virus strain, and provide a measure of the activation of the host immune response.

A general limitation to using RNA-based tests, however, is that accurately measuring blood RNA biomarkers in a clinical setting has been hampered by the instability of blood RNA and the complexity of RNA purification and quantitation. Thus, RNA biomarkers, while sensitive and accurate, are presently too slow to be applied in any acute clinical setting. Conversely, using circulating protein biomarkers of infection or sepsis is technically more feasible in the clinic, but sensitivity and accuracy of most current measures are marginal. While circulating markers, such as C-reactive protein (CRP) and lactate, are currently in clinical use, they have relatively weak predictive power relative to potentially life-threatening outcomes such as sepsis [4]. While many promising markers have been identified, their use in the clinic have been hampered by the complexity of blood and its innate ability to bind or degrade inflammatory cell products.

While COVID-19 is principally initiated and driven by SARS-CoV2 virus, next generation sequencing proves that secondary bacterial and fungal infections are readily detected in COVID-19 cases. Secondary infections in COVID-19 patients have been detected in the range of 25–50% of patients, with a predominance of Klebsiella and Staphylococcus [5, 6]. The severity of COVID-19 syndrome has been consistently shown to be related to an increase in the neutrophil/lymphocyte ratio (NLR) [7, 8]. Severe COVID-19 is thought to result from a 'cytokine storm' devolving into intrapulmonary and intravascular neutrophil extracellular traps (NETs) that compromise oxygen exchange [9–12]. Thus, while neutrophil activation might represent a useful strategy for detecting infection and the host response, a practical method at the point of care is lacking. Furthermore, purification of neutrophils from human blood by conventional density methods for RNA or functional analysis is time-consuming and typically requires centrifugation that can yield activated or damaged cells.

Thus, a powerful addition to the diagnostic arsenal would be a test that detects novel or known pathogens through the activation of the host immune response. The present studies employed RNA sequencing (RNAseq) to identify blood RNA biomarkers of COVID-19 infection and severity, and suggested the use of NET-related factors as alternative biomarkers that could be measured rapidly and inexpensively at the point of care.

## Materials and methods

### Enrollment and blood draw

This prospective, observational study was approved by the Institutional Review Board of The George Washington University (#NCR202539). All subjects, or their legally authorized surrogate, provided informed consent. Patients admitted into The George Washington University Hospital Intensive Care Unit were screened for inclusion and exclusion criteria. Inclusion criteria consisted of age greater than or equal to 18 and the presence of a positive SARS-CoV2 test. Normal controls were sampled from asymptomatic medical center personnel known or assumed to be SARS-CoV2 negative. After informed consent was received, venipuncture, or sampling from an indwelling catheter, was performed to draw whole blood in Tempus Blood RNA tubes (ThermoFisher) and BD Vacutainer K2 EDTA tubes (BD Biosciences). Researchers who were processing the blood samples and performing assays were blinded to the subjects' clinical characteristics and outcomes by anonymous coding of the samples.

### Neutrophil isolation with anti-CD15+ magnetic beads

Within 20 minutes post-collection, the CD15+ cells (comprising neutrophils, scarce eosinophils, and a rare subtype of T cells) were isolated from the EDTA-treated blood sample using anti-human CD15 antibody-coated Dynabeads (Invitrogen) with minor modifications to the manufacturer's protocol. Immediately before use, the beads were re-suspended and washed as recommended. The bead isolation buffer (BIB) was modified by using HBSS without calcium or magnesium (Gibco). Blood volumes of 750 μl were used in 1.5 mL tubes. When larger volumes were processed, multiples of 750 μl were used. Instead of the recommended initial wash steps, the blood was spun at 3000 rpm (604 g) for 10 minutes, the plasma removed, and 750 μl of BIB was added just before adding the beads. Each wash step used a 1:1 volume to the original blood (typically 750 μl BIB), to keep a 1x concentration of the CD15+ cells. Cell capture and washing were performed using the MAGNA-SEP magnet (Invitrogen) for two minutes. Cell counts of the 1x isolate were performed using 10 μl in a hemocytometer at 200X using phase contrast. The identities of the isolated cells were confirmed by fixing with 4% formaldehyde in 1x PBS, and using a May-Grünwald stain (Wright-Giemsa stain with eosin Y).

### Whole blood cell count and CD15+ yield calculation

For both white blood cell counts (WBC) and differential counts, we used the same blood as for the isolation of the CD15+ cells and they were performed simultaneously with the isolation. WBC counts were achieved by adding 20 μl of whole blood to a Leukotic tube (Bioanalytic), following the product's protocol, ensuring the cells remained homogenously suspended, and counting the preserved cells within 2 hours in a hemocytometer at 200X. Whole blood smears were performed in triplicate. Differential staining was achieved by fixation in methanol (10 min; ThermoFisher Scientific), eosin Y staining (40 s; Sigma Aldrich), Wright Giemsa staining (100% 1:30 min, then 50% 3 min; VWR), and finishing with Permount (Electron Microscopy Sciences) and a coverslip. For each sample, 200 or more cells were counted within the feathered edges of the smears. CD15+ yield calculations were performed by comparing the isolate's

cell count to the WBC's fraction of neutrophils and eosinophils as determined by the differential count.

## Whole blood RNA extraction

The Tempus Blood RNA preservation tubes (Applied Biosystems) were stored at -80° C and thawed in a 37° C water bath. RNA was purified using the Tempus Spin RNA Isolation Reagent Kit (Applied Biosystems). The Tempus blood lysate was spun at 3000 g for 30 minutes, and the pellet was then resuspended and transferred onto purification columns. Following on-column DNAse treatment using a TurboDNase kit (Ambion) and a series of washing and drying steps, purified RNA was eluted with nuclease-free water. RNA levels were quantified using the NanoDrop ND-1000 Spectrophotometer and quality was determined by capillary electrophoresis on the Agilent 2100 Bioanalyzer.

## RNA biomarker digital droplet PCR (ddPCR)

Primers and probes for DEFA1, IL8RB, ALPL, RSTN, MPO, and ACTB were designed using Geneious Prime software and were synthesized by Biosynthesis. RNA (200 ng) was reverse transcribed into cDNA using gene-specific primers (250 nM) with the RNAse H+ reverse transcriptase and 5X Supermix from the iScript Select cDNA Synthesis Kit (Bio Rad). The cDNA was diluted 1/20 in nuclease-free water and placed in a ddPCR 96-well plate (5 μl per well). Each of the markers were paired to make a total of three primer-probe master mixes (4 μM forward primer; 4 μM reverse primer, 1.12 μM probe) with water and the ddPCR Supermix (Bio Rad); 17 μl were added per ddPCR well.

The BioRad Automated Droplet Generator produced at least 10,000 nanoliter-sized droplets per well, the target transcripts were amplified in each droplet by PCR, and each droplet was read separately by the Bio Rad QX200 Droplet Reader. Bio Rad Quantasoft software was used to set the threshold for positive droplets. Copy numbers were normalized by dividing by the number of ACTB copies; this was performed to counter variation in the numbers of cells captured and possible variation in RNA yield efficiency.

## SARS-CoV2 nucleocapsid RNA ddPCR

The Bio-Rad SARS-CoV-2 ddPCR Kit (FDA-cleared for COVID diagnosis from nasal swab RNA) was used to measure nucleocapsid transcripts (N1 and N2) of the SARS-CoV2 in Tempus whole blood RNA. cDNA synthesis and ddPCR was accomplished with a one-step RT-ddPCR mix (Bio Rad) added directly to the RNA. A mix of primers and FAM/HEX probes for N1, N2 and human RNase P (RP) was added (Bio Rad). After droplet generation, the thermal cycler reverse transcribed the RNA and then amplified the cDNA. Droplets were read using the Bio Rad QX200 and clusters were gated using the Bio Rad Quantasoft Analysis Pro software. RP was included as a control to confirm proper amplification and reading. Samples generating two or more N1+ or N2+ droplets were considered positive as specified by the manufacturer.

## RNAseq methods

Whole blood RNA was extracted and DNAsed using the Tempus system described above. Quantity and quality were assessed using a NanoDrop for OD at 260/280 and an Agilent 2100 Bioanalyzer. Each sample had a RIN >8. The RNA (200 ng) was depleted of ribosomal RNA, converted to cDNA, tagged for multiplexing, and amplified using the Illumina TruSeq Stranded Total RNAseq kit. After quality control adjustments, the multiplexed cDNA library

was sequenced on an Illumina NextSeq using 24 samples per run with the Illumina TotalRNA labeling and amplification kit.

**Alignment and transcript counting.**   The raw files containing 150-bp paired-end reads were concatenated across the 4 sequencing lanes, quality-filtered and trimmed using trimmomatic in the Galaxy software suite. Trimmed reads were aligned to the human hg38 genome (UCSC 197K transcripts) using HiSat2, and then parsed into known transcripts using feature-Count. The average yield of aligned paired-end reads was 12.4 million per sample.

**Bioinformatic analysis.**   The raw read counts per transcript and transcript sizes were transferred to Excel, where the RPKM was computed (reads per thousand bases of transcript, per million total reads, per patient). The RPKM per transcript per patient was loaded into GeneSpring GX14 to analyze for differentially expressed genes (DEGs). The 197K transcripts were reduced to the gene level (~44K transcripts) and then the DEGs were identified by a triple-filter strategy that excludes transcripts of low expression (<0.01 RPKM in >70% of samples), and then seeks transcripts with a greater than 2-fold change in the geometric mean expression with a t-test p-value of <0.001 (uncorrected for multiple testing). This approach has been shown to be accurate in identifying spiked 'true' DEGs in prior studies [13]. Gene ontologies were analyzed with NIH David, and pathways analyzed with Ingenuity Pathway Analysis.

**Cell type-specific gene expression.**   Pre-curated transcripts from the Human Blood Atlas identified by RNAseq of purified blood cell types [14] were matched to gene-level transcripts in the RNAseq data set. The average level of expression among detectable transcripts (non-zero in at least one group) was computed and used as an index of cell-type specific expression.

## Elastase assay

CD15+ neutrophils for the fluorometric elastase assay were isolated as described above. Aliquots of 50 μl containing 100-600K cells were stored at -80˚ C. The elastase assay was performed in batches of 4–16 samples. Each batch was thawed and underwent 5 cycles of freeze/thaw using dry ice in ethanol and a 37˚ C water bath. Thawed lysates were diluted with 140 μl of PBS without calcium or magnesium and 10 μl of the fluorometric substrate, bis(N-benzyloxycarbonyl-L-tetra-alanyl)rhodamine, (CBZ-Ala-Ala-Ala-Ala)2Rh110, (Anaspec), dissolved in 90% DMSO to 2000 μM. Proteolytic cleavage of the substrate de-quenches the Rhodamine 110 fluorochrome. At specified times, the magnetic beads were removed from the light path with a strong neodymium magnet, and fluorescence intensity was measured on a Qubit 2.0 fluorometer (ThermoFisher Scientific) with 470 nm excitation and 520 nm emission. Mixing and measurements were repeated every 20 minutes for 2 hours.

# Results and discussion

## Clinical characteristics of COVID-19 patients in ICU

Patients were consented and blood was drawn upon admission to the intensive care unit (ICU) near the peak of the COVID-19 pandemic, from November 2020 through March 2021. Because the ICU was used to isolate any suspected COVID-19 cases, they reflected a range of cases from PCR+ but asymptomatic, through moderate to severe COVID-19 symptoms (n = 38). For biomarker analysis, they were divided into 3 groups: 1) Incidental patients (Incidental, n = 7), were SARS-CoV2 PCR+ in nasal swabs, but their COVID-like symptoms were mild or unnoticeable, and were not the reason for admission to the hospital/ICU. These patients were admitted for a range of reasons including trauma and elective surgery. 2) Moderate COVID patients (Moderate, n = 7) were admitted for symptoms consistent with COVID-19 and tested PCR+ by nasal swab, but did not require intubation or vasopressors. 3) Severe

COVID-19 patients were PCR+ by nasal swab and met at least 1 of 3 criteria: intubation, vaso-pressors, or fatality (Critical, n = 24).

The patient characteristics are consistent with prior observations that more severe cases of COVID-19 are associated with increasing age and BMI (Table 1). There were not striking associations of severity with comorbid conditions such as Type II diabetes or hypertension (Table 1), or other comorbidities such as cardiovascular or pulmonary disease, although it is important to note that the present study was not designed, nor powered to detect those differences (S1 Table). Of note, there was a considerable discrepancy between the clinical classification of the patients, based on nasal swab PCR for SARS-CoV2 and symptoms, versus the laboratory measures of SARS-CoV2 viral titer in blood as determined by ddPCR. In short, 2 of the 7 Incidental patients, and one of the Moderate patients actually had very high titers of SARS-CoV2 in blood, which obscured the predicted differences in blood titers between groups (Table 1).

## RNA sequencing of blood RNA from COVID-19 patients

To gather a relatively unbiased, transcriptome-wide view of the immune response to SARS--CoV2, Tempus-preserved whole blood RNA was subjected to RNAseq. Prior microarray-

**Table 1. Clinical characteristics of ICU patients included in the study.**

|  | Incidental |  | Moderate |  | Critical |  |  |
|---|---|---|---|---|---|---|---|
| **Demographics** | **Mean** | **S.E.M.** | **Mean** | **S.E.M.** | **Mean** | **S.E.M.** |  |
| **N per group** | 7 |  | 7 |  | 24 |  |  |
| **Sex (% Male)** | 57.1% |  | 42.9% |  | 45.8% |  |  |
| **Age (years)** | 54.0 | 6.9 | 46.7 | 3.2 | 62.1 | 1.7 | * |
| **BMI** | 30.4 | 2.7 | 37.3 | 4.1 | 32.8 | 1.9 |  |
| **African or African-American** | 42.9% |  | 85.7% |  | 58.3% |  |  |
| **Co-Morbidity** |  |  |  |  |  |  |  |
| **T2DM (Type II Diabetes Mellitus)** | 42.9% |  | 42.9% |  | 50.0% |  |  |
| **HTN (Hypertension)** | 71.4% |  | 57.1% |  | 66.7% |  |  |
| **Treatments** |  |  |  |  |  |  |  |
| **Intubation** | 14.3% |  | 0.0% |  | 79.2% |  | * |
| **BiPAP** | 0.0% |  | 42.9% |  | 12.5% |  |  |
| **ECMO VV** | 0.0% |  | 0.0% |  | 8.3% |  |  |
| **Dexamethasone** | 57.1% |  | 100.0% |  | 83.3% |  |  |
| **Vasopressor** | 0.0% |  | 0.0% |  | 66.7% |  | * |
| **Clinical Measures** |  |  |  |  |  |  |  |
| **PF Ratio** | 2.51 | 0.95 | 1.41 | 0.42 | 1.76 | 0.26 |  |
| **Fatality** | 14.3% |  | 0.0% |  | 50.0% |  | * |
| **Laboratory Results** |  |  |  |  |  |  |  |
| **SARS-CoV2 blood titer (ddPCR)** | 1068.77 | 969.39 | 1063.36 | 1094.9 | 478.3 | 272.63 |  |
| **WBC Count** | 11.99 | 3.43 | 10.03 | 2.43 | 14.98 | 1.37 |  |
| **Lymphocyte Count** | 1.18 | 0.36 | 1.13 | 0.26 | 1.05 | 0.17 |  |
| **Neutrophil Count** | 9.65 | 2.23 | 9.88 | 2.41 | 14.05 | 1.32 |  |
| **Neutrophil/Lymphocyte Ratio (NLR)** | 10.23 | 2.88 | 11.93 | 5.13 | 25.38 | 4.69 |  |
| **Platelet Count** | 308.71 | 65.9 | 223.14 | 37.05 | 273.82 | 26.19 |  |
| **Hemoglobin** | 11.13 | 1.17 | 12.39 | 0.99 | 9.38 | 0.47 | * |
| **Yield (ug) from 3 mL blood** | 9.3 | 3.1 | 10.8 | 2.8 | 15.0 | 2.4 |  |

based profiling in the large MOSIAC trial identified a progression from interferon-induced to neutrophil-associated transcriptome changes in Tempus-preserved whole blood of patients with severe influenza [15]. For the current RNAseq analysis (n = 24), a subset of normal controls (n = 4) was combined with Incidental COVID patients (n = 3), and compared with combined Moderate and Critical COVID cases (n = 17). The results indicate that of the 44,783 total transcripts quantified at the gene level, 758 DEGs were identified between CON and COVIDs (S2 Table). Of these 758 DEGs, 603 transcripts were decreased in the COVID cases, while 155 transcripts were increased (4:1 decreased/increased ratio). Representative DEGs with their associated fold-changes are shown in Fig 1 (Panel A). The COVID-19 patients showed striking increases of >20-fold in known neutrophil-related transcripts such as Defensin alpha1 (DEFA1), matrix metallopeptidase 8 (MMP8), and myeloperoxidase (MPO). At least 67 of the 603 decreased transcripts were related to isoforms of the T-cell receptor. Also decreased were CD4, GATA3, ZAP70, TIGIT, IKZF2, ICOS and CD3D, intimately related to T cell function. The decreases in T-cell related transcripts are less striking in magnitude (3–5 fold) than the increased transcripts (20–25 fold), but T-cells (~5% of WBC) are a smaller subset of cells relative to neutrophils (>40% of WBC).

Gene ontology analysis indicated that the increased transcripts were most strongly associated with the innate immune response, with numerous additional associations to the immune response, especially defense against microorganisms (S2B Table). Transcripts decreased by COVID-19 were mapped to transcription and translation, with prominent associations to the adaptive immune response, T cells and TOR signalling (S2C Table). Pathway analysis indicated strong overlap with precurated pathways relevant to T helper cell differentiation and signaling, Th17 activation, T cell apoptosis and exhaustion (Fig 1, Panel B). Careful manual curation indicated that among the 155 increased transcripts, many were strongly associated with neutrophils or B-cells, while many of the 603 decreased transcripts were strongly specific to T-cells or NK cells.

## COVID-19 markedly alters cell-type-specific transcripts

To determine whether this cell-specific effect was due to an interpretation bias, the transcriptomes were compared to cell-type-specific transcriptomes from the Human Blood Atlas. The Blood Atlas' data has been previously used to demonstrate enrichment in specific blood cell types in RNAseq samples [14], and was able to detect changes in T-cell abundance in coronary artery disease [16]. Average levels of the cell-type-specific transcripts are shown for control/incidental and critical patients (Fig 1, Panel C). Again, there is a relative increase in neutrophil- and B-cell-related transcripts and a decrease in T-cell related transcripts. Notably, the average increase in pre-curated neutrophil-specific transcripts (~3-fold, Panel C) is far less than the increase of some of the neutrophil-related DEGs (~25-fold, Panel A), suggesting that increased neutrophil counts alone do not fully explain the increased transcript levels. Rather, a plausible explanation is that transcript levels change during neutrophil "activation", a broad term for the complex processes that lead to NETosis (neutrophil extracellular trap formation) and autolysis in response to various pathogen-related or environment-related stimuli.

Thus, the RNAseq provides very specific RNA-level evidence for the NLR imbalance that has been reported previously using more conventional cell counting methods. However, it is also possible that contributions of other cell types to these transcripts are relevant.

## Biomarker identification for COVID-19 syndrome

Among the transcripts significantly increased by COVID-19 were ALPL, DEFA1, DEFA3, and MPO, which we had previously identified in non-COVID infections, such as pneumonia and

## A) Differentially Expressed Transcripts

| Fold Change | Gene Symbol | Transcript Description | Cell Type |
|---|---|---|---|
| **INCREASED IN COVIDs** | | **representing 155 total DEG** | |
| 3.45 | ALPL | alkaline phosphatase, liver/bone/kidney | neutrophil |
| 9.90 | BPI | Bactericidal Permeability Increasing Protein | neutrophil |
| 13.41 | CTSG | cathepsin G | neutrophil |
| 20.00 | DEFA1 | Defensin 1 | neutrophil |
| 25.80 | DEFA3 | defensin, alpha 3, neutrophil-specific | neutrophil |
| 20.53 | DEFA4 | defensin alpha 4 | neutrophil |
| 5.87 | IG*** | Immunoglobin transcripts (12 observed) | B cell |
| 8.26 | IL1R2 | interleukin 1 receptor type 2 (decoy receptor) | multiple |
| 33.34 | MMP8 | matrix metallopeptidase 8 | neutrophil |
| 7.46 | MMP9 | matrix metallopeptidase 9 | neutrophil |
| 16.12 | MPO | myeloperoxidase | neutrophil |
| | | | |
| **DECREASED IN COVIDs** | | **representing 603 total DEG** | |
| -3.41 | CD28 | CD28 molecule, co-stimulatory | T cell |
| -3.91 | CD3D | CD3d molecule, T cell receptor Delta chain | T cell |
| -4.52 | CD4 | CD4 molecule | T cell |
| -2.78 | GATA3 | GATA binding protein 3 | T cell |
| -2.73 | ICOS | inducible T-cell co-stimulator (ICOS) | T cell |
| -3.43 | IKZF2 | IKAROS family zinc finger 2, aka Helios | lymphocytes |
| -5.37 | KLRB* | killer cell lectin like receptors (8 observed) | NK cells |
| -2.52 | RPS/L* | ribosomal proteins (32 observed) | multiple |
| -3.50 | TCF7 | transcription factor 7 (T-cell specific, HMG-box) | T cell |
| -3.71 | TIGIT | T cell immunoreceptor with Ig and ITIM domains | T cell |
| -3.71 | TR** | T cell receptors (67 observed) | T cell |
| -3.51 | ZAP70 | zeta chain of T cell receptor kinase 70kDa | T cell |

## B) Top Canonical Pathways

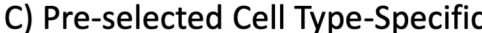

| Pathway | % Overlap | (Hits/Total) | p-value |
|---|---|---|---|
| **T Helper Cell Differentiation** | 15.9% | (75/471) | 5.8E-37 |
| **ICOS-ICOSLG Signaling in T Helper Cells** | 15.2% | (77/507) | 1.8E-36 |
| **Th17 Activation Pathway** | 15.5% | (75/484) | 3.9E-36 |
| **Calcium-induced T Lymphocyte Apoptosis** | 15.7% | (72/460) | 5.5E-35 |
| **T Cell Exhaustion Signalling Pathway** | 13.8% | (78/566) | 6.5E-34 |

## C) Pre-selected Cell Type-Specific

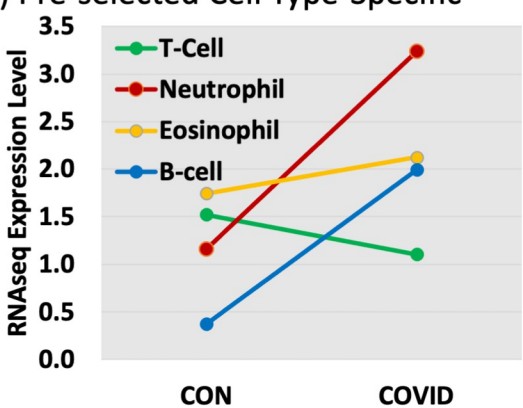

**Fig 1. Cell type-specific and differentially expressed genes (DEGs) between COVID-19 and controls.** *Panel A:* The RNAseq data was analyzed to identify differentially expressed genes (DEGs) between controls/incidentals (7) and COVID-19 patients (17) using a triple filtering approach that excluded transcripts with low absolute levels (<0.01 RPKM), and then testing for an absolute fold-change of >2 with a t-test p-value of less than 0.001 uncorrected. A representative subset is shown for increased (red) vs decreased (green) transcripts in COVID-19 patients. *Panel B:* The total list of 758 DEGs was analysed for the pathways affected using IPA. The 5 top pathways are shown with the % of overlap to the precurated list, the number of transcript matches, and the p-value of the overlap. *Panel C:* Cell-type-specific RNAs (~10 per cell type) were extracted from the Blood Atlas and then their levels were computed from the RNAseq data of control vs COVID-19 patients. The points reflect the mean expression level of the transcripts in RPKM.

acute appendicitis. Also increased by COVID was the transcript for cathepsin G, a well-known neutrophil elastase with activity against viruses and bacteria. Cathepsin G (CTSG) was elevated 13-fold in COVIDs over controls (p<$10^{-5}$). Neutrophils use 2 other elastases, neutrophil elastase (ELANE) and proteinase 3 (PR3, PRTN3), but their RNA levels were only detectable in some COVID patients. By comparison to several other studies from our lab and others, the magnitude of the increases in the neutrophil-related transcripts is quite strong, ranging from >3-fold for ALPL to >30-fold for MMP8, with substantial changes in the neutrophil defensins (20-25-fold) and myeloperoxidase (16-fold).

## RNAseq identifies biomarkers of COVID-19 severity

**Severity defined by SARS-CoV2 seropositivity.** The prior analysis compared controls to COVID-19 cases defined by clinical measures and SARS-CoV2 positivity in nasal swabs. However, the clinical severity of COVID-19 is affected by many factors that are independent of viral infectivity, especially pre-existing, comorbid conditions. Thus, to partially isolate the SARS-CoV2 contribution to disease severity, the controls were excluded, and the 20 COVID-19 patients were grouped according to seropositivity for SARS-CoV2 RNA in stabilized whole blood. Using a similar fold-change/p-value (>2-fold, <0.005) filtering of seropositive (n = 10) vs seronegative (n = 10) cases, 38 increased transcripts and 68 decreased transcripts were identified (S3A Table). A manually curated list of selected transcripts with useful annotations is shown in Table 2, and reinforces the conclusion that increased transcripts tend to be

neutrophil/innate immunity related, while decreased transcripts are related to adaptive immunity, especially T cells.

## Severity defined by vasopressor use

ICU patients with severe COVID-19 infections can become hypotensive, and thus are often given vasopressors, such as epinephrine and/or angiotensin II. Using vasopressor administration as an indicator of COVID-19 severity, the 24 critical patients were subdivided into Vasopressor (n = 9) and No-Vasopressor (n = 15) groups, and then reanalyzed for DEG at 2-fold change and p<0.05 uncorrected p-value. The results identified 33 DEG at the gene level, of which 11 were increased and 22 were decreased (S4 Table). The results echoed the overall results in the respect that COVID-19 patients on vasopressors had increased expression of neutrophil-like markers (BPI, CAMP, DEFA3) and decreased expression of T-cell related markers such as T-cell receptor alpha joining 50 (TRAJ50) (Fig 2).

**Severity defined by survival.** The critical COVID-19 patients were alternatively subdivided according to their survival, which identified 476 DEGs (S5 Table) and a similar trend was observed (Fig 2, lower panel). Patients that had a fatal outcome in the ICU (n = 5), when compared to survivors (n = 19), had drastically higher levels of RNAs for the neutrophil-related transcripts such as MMP8 (14-fold), MPO (5-fold) and DEFA1 or DEFA3 (8-fold). Conversely, the fatal COVID-19 patients exhibited 2–4 fold lower levels of T cell-related transcripts such as CD4, CXCR3, CXCR5, CXCR6, IKZF2/3, and T cell receptor alpha joining 52 (TRAJ52).

Notably, the patients that succumbed to COVID-19 displayed 2-fold lower levels of the CXCR6 transcript (Fig 2, lower panel), a memory T follicular helper (Tfh) marker, which was recently associated with COVID-19 severity by an expression quantitative trait loci (eQTL) analysis [17]. Furthermore, the current RNAseq results confirm ~30% of the transcripts identified by other labs using single cell RNAseq of PBMC of 6 mild vs 11 severe COVID-19 patients, much greater than expected by chance (see S6 Table for 22 near or exact matches, 18-fold enrichment, p = 6.3 x $10^{-22}$) [18].

Thus, a consistent trend emerges that the more severe the COVID-19, the more elevated are the neutrophil RNAs and more decreased are the T cell-related transcripts. This trend is consistent with previously published reports suggesting that the neutrophil/lymphocyte ratio (NLR) is one of the best correlates of COVID-19 disease severity [19–21].

## Blood RNA analysis by ddPCR confirms biomarkers in COVID-19 patients

The RNAseq analysis above, and prior blood transcriptome studies of patients with pneumonia or appendicitis, demonstrated elevated levels of five neutrophil-related RNAs: defensin a1 (DEFA1), alkaline phosphatase (ALPL), interleukin-8 receptor beta (IL8RB or CXCR2), myeloperoxidase (MPO) and resistin (RSTN). Their elevation was confirmed in a larger group of COVID patients using an independent method of RNA quantification.

In the larger set of 38 patients from the ICU, the RNA biomarkers were quantified using droplet digital PCR (ddPCR) with gene-specific primers for cDNA synthesis. The patients were stratified by COVID symptom severity into incidental, moderate and critical COVID groups similar to the RNAseq study. Using conventional WBC and differential counts, there was a clear trend toward elevated WBC and neutrophil counts, decreased lymphocyte counts, and increased neutrophil/lymphocyte ratios (NLR), but these measures failed to differentiate between the severity groups with statistical significance (Fig 3). Likewise, lactate, CRP, and creatinine were not statistically different between groups. The RNA measures, however, showed striking and statistically significant differences between groups, with DEFA1, in particular,

**Table 2. Selected transcripts differentially expressed in SARS-COV2 seropositive vs seronegative cases.**

| GeneSym | Description | SARS(-) | SARS(+) | FOLD | Putative Function in COVID-19 |
|---|---|---|---|---|---|
| **Transcripts INCREASED in RNAemia (38 total)** | | N = 10 | N = 10 | **CHANGE** | |
| ASPH | aspartate beta-hydroxylase | 12.09 | 31.42 | 2.60 | induces epitope-specific T cell responses |
| C5orf30 | chromosome 5 open reading frame 30 | 3.79 | 14.69 | 3.88 | autoimmune susceptibility, mphage resolution of inflammation |
| CDK5R1 | cyclin-dependent kinase 5, regulatory subunit 1 (p35) | 2.69 | 6.47 | 2.41 | TGF-B sensitizes TRPV1 thru CDK5 signaling |
| CLEC4E | C-type lectin domain family 4 member E | 73.86 | 167.84 | 2.27 | damage associated molecular pattern DAMP |
| DAAM2 | dishevelled associated activator of morphogenesis 2 | 0.80 | 7.54 | 9.41 | formin protein, hypoxia, DAAM1 increased in SARS-ferret model |
| DGKH | diacylglycerol kinase eta | 1.56 | 3.53 | 2.26 | GWAS link to neutrophil abundance |
| ERLIN1 | ER lipid raft associated 1 | 5.75 | 11.65 | 2.03 | upregulated in sepsis, neutrophils |
| FKBP5 | FK506 binding protein 5 | 27.28 | 76.85 | 2.82 | regulates Influenza A infection with RIG-I |
| FLT3 | fms related tyrosine kinase 3 | 3.38 | 12.21 | 3.62 | related to COVID anticoagulation |
| GPR160 | G protein-coupled receptor 160 | 7.23 | 15.67 | 2.17 | regulates mycobacteria entry into macrophages |
| HMGB2 | high mobility group box 2 | 102.94 | 223.43 | 2.17 | VdJ recombination, mediates innate immunity to viral infection |
| INHBB | inhibin beta B | 0.10 | 0.90 | 9.01 | activin B, TGF-B axis |
| JDP2 | Jun dimerization protein 2 | 8.09 | 17.90 | 2.21 | GWAS to scrub typhus susceptibility, neutrophil |
| KCNE1 | potassium voltage-gated channel subfamily E 1 | 1.99 | 4.96 | 2.49 | novel target for immunomodulation in leukocytes |
| LINC00659 | long intergenic non-protein coding RNA 659 | 0.10 | 1.36 | 13.56 | high altitude thrombosis by spongeing mir-143, -15 |
| OLAH | oleoyl-ACP hydrolase | 2.64 | 21.57 | 8.16 | increased in Influenza infection |
| PHC2 | polyhomeotic homolog 2 | 45.55 | 97.97 | 2.15 | detected in immune cells of RA |
| POLQ | polymerase (DNA) theta | 0.73 | 1.82 | 2.50 | somatic hypermutation of Ig genes |
| SAP30 | Sin3A-associated protein | 4.66 | 13.96 | 3.00 | interacts with NS protein of Rift Valley Fever virus |
| SCRG1 | stimulator of chondrogenesis 1 | 0.20 | 1.36 | 6.67 | decreases tristetraprolin, autophagy related |
| SLC5A9 | solute carrier family 5 member 9 | 0.10 | 0.97 | 9.72 | SGLT4, mannose and fructose transporter |
| STS | steroid sulfatase (microsomal), isozyme S | 1.99 | 4.14 | 2.08 | interferon-gamma induced |
| TPST1 | tyrosylprotein sulfotransferase 1 | 9.16 | 30.04 | 3.28 | macrophage innate immune stimulated by TLR ligands |
| TSC22D3 | TSC22 domain family member 3 | 108.24 | 263.41 | 2.43 | related to T cell dysfunction |
| VSIG4 | V-set and immunoglobulin domain containing 4 | 5.28 | 18.11 | 3.43 | VSIG+ Macs suppress T cell proliferation, suppresses TLR4 path |
| ZNF608 | zinc finger protein 608 | 2.43 | 15.92 | 6.56 | increased in SARS-CoV2 Ferret model, B-cell |
| **Transcripts DECREASED in RNAemia (68 total)** | | | | | |
| A2M | alpha-2-macroglobulin | 4.19 | 0.93 | 4.52 | TGF-B scavenger |
| ACSM3 | acyl-CoA synthetase medium-chain family member 3 | 2.56 | 1.04 | 2.46 | blood pressure control |
| ANK3 | ankyrin 3 | 2.12 | 0.75 | 2.84 | T reg cells |
| CCR3 | Receptor for a C-C type chemokine. | 6.01 | 0.60 | 10.08 | Th2 associated chemokine receptor |
| CD244 | CD244 molecule | 9.54 | 3.67 | 2.60 | CD8 T and NK cytotoxicity |
| CNR2 | cannabinoid receptor 2 | 1.88 | 0.31 | 6.14 | macrophage related |
| CYSLTR2 | Receptor for cysteinyl leukotrienes. | 2.98 | 0.25 | 11.81 | Leukotriene receptor |
| DZIP3 | DAZ interacting zinc finger protein 3 | 3.00 | 1.26 | 2.38 | induced by RSV infection |
| EPHA4 | EPH receptor A4 | 2.42 | 0.63 | 3.85 | activation of EPHA4 on CD4+CD45RO+ memory cells |
| FASLG | Fas ligand transcript variant 1 | 3.10 | 0.50 | 6.15 | major regulator of apoptosis |
| GNLY | granulysin | 75.96 | 24.73 | 3.07 | stimulated CD8+ T cells |
| IL11RA | interleukin 11 receptor, alpha | 3.79 | 1.76 | 2.15 | T cell |
| IL5RA | interleukin 5 receptor subunit alpha | 5.51 | 0.22 | 25.28 | overexpressed in asthma, dermatitis, rhinitis |
| INPP4B | inositol polyphosphate-4-phosphatase type II B | 7.48 | 2.97 | 2.52 | corrlelates to testosterone and immune response in COVID19 |
| LINC00612 | long intergenic non-protein coding RNA 612 | 1.78 | 0.21 | 8.40 | regulates Notch1 apoptosis, inflammation, and ox stress |
| LINC00861 | long intergenic non-protein coding RNA 861 | 11.24 | 4.27 | 2.63 | implicated in sepsis |
| LTBP4 | latent transforming growth factor beta binding 4 | 2.11 | 0.67 | 3.15 | TGF-B pathway critical for T reg differentiation |

*(Continued)*

**Table 2.** (Continued)

| GeneSym | Description | SARS(-) | SARS(+) | FOLD | Putative Function in COVID-19 |
|---------|-------------|---------|---------|------|-------------------------------|
| MAF | v-maf oncogene homolog | 5.06 | 1.85 | 2.73 | antiviral defense against Hep B virus |
| MIR151B | microRNA 151b | 3.86 | 0.68 | 5.66 | discriminates T from B cell lymphoma |
| MYBL1 | MYB proto-oncogene like 1 | 4.81 | 1.35 | 3.57 | T cell dependent B cell response |
| PZP | pregnancy-zone protein | 1.69 | 0.10 | 16.92 | biomarker of disease recovery COVID19 |
| RNF182 | ring finger protein 182 | 24.02 | 3.44 | 6.99 | inhibits TLR-triggered cytokine production |
| RORA | RAR related orphan receptor A | 7.71 | 3.12 | 2.47 | T cell differentiation |
| TNFRSF25 | tumor necrosis factor receptor superfamily, 25 | 5.01 | 1.86 | 2.69 | aka DR3, costimulatory for T reg |
| TRAF1 | TNF receptor associated factor 1 | 6.45 | 2.40 | 2.69 | T and B cell |
| TR*** | T Cell Receptor Associated (18 transcripts) | 10.65 | 3.96 | 2.69 | T cell receptor |
| UBASH3A | ubiquitin associated and SH3 domain containing A | 5.40 | 1.69 | 3.19 | T cell activation, negative regulator of activation |
| YBX1 | Y-box binding protein 1 | 322.09 | 157.79 | 2.04 | antibacterial response |

showing stepwise increases from 6.37% of actin B (%ACTB) in normal controls, to 19.7% in Incidental COVIDs, 62.5% in Moderate COVIDs, and 86.2% in Critical COVIDs (Fig 3, lower left panel). ALPL and IL8RB RNA levels, which we have shown are related to biofilm-type infections, such as appendicitis, also showed detectable increases. While expressed at overall lower absolute values than DEFA1, myeloperoxidase (MPO) and resistin (RSTN) RNAs showed strong and significant changes in the COVID-19 patients (Fig 3, lower right panels).

The DEFA1 score in whole blood was modestly correlated with the Sequential Organ Failure Assessment (SOFA) score as an indicator of COVID-19 severity (Pearson R = 0.45, Fig 4). The SOFA score estimates the severity and number of organ systems involved in the disease. This analysis identified 3 patients with low to medium SOFA scores (6–11 on a scale of 24) that had apparently normal DEFA1 scores (<10%) that were classified as 'Critical'. However, their viremia was at or below the threshold of detection of N1 or N2 spike protein transcripts by ddPCR, suggesting their illness may originate from non-SARS causes. Further, 2 of the 3 patients had elevated 'biofilm' biomarkers (APLP+IL8RB) suggesting they had non-SARS-CoV2 co-infections. The APACHE2 score was also positively, but slightly less correlated to DEFA1 score (Pearson R = 0.31, not shown).

## Sensitivity and specificity of DEFA1 RNA for detecting SARS seropositivity

Using whole blood RNA, SARS-CoV2 viremia (RNAemia) was measured by Bio-Rad ddPCR for 2 regions of the nucleocapsid protein in each of the whole blood RNA samples. Prior studies indicated that the presence of virus in plasma is associated with a worse prognosis [22, 23]. The DEFA1 score was stratified into 5 categories (1 = 0–5%, 2 = 6–10%, 3 = 11–20%, 4 = 21–50%, 5 = >51%) and compared to a binary classification of SARS-CoV2 using the specified thresholds for the ddPCR test. The results indicate that DEFA1 was 95.5% sensitive in detecting SARS-CoV2 infection, with 41.2% specificity, for overall 71.1% accuracy (ROC area = 0.697). However, it is important to note that many of SARS-CoV2 negative patients were known to have had other infections that could have triggered DEFA1 activation, and thus, the 41.2% specificity is unrealistically low. Only 1 of 21 'true' positives was missed.

## Whole blood vs. purified neutrophil levels of RNA biomarkers of infection

Because DEFA1, in particular, is considered a neutrophil defensin, it was hypothesized that the DEFA1 signal would be highly enriched in isolated CD15+ cells, including >70% of the whole blood's neutrophils. Thus, in a subset of 23 patients, the total blood RNA, the CD15

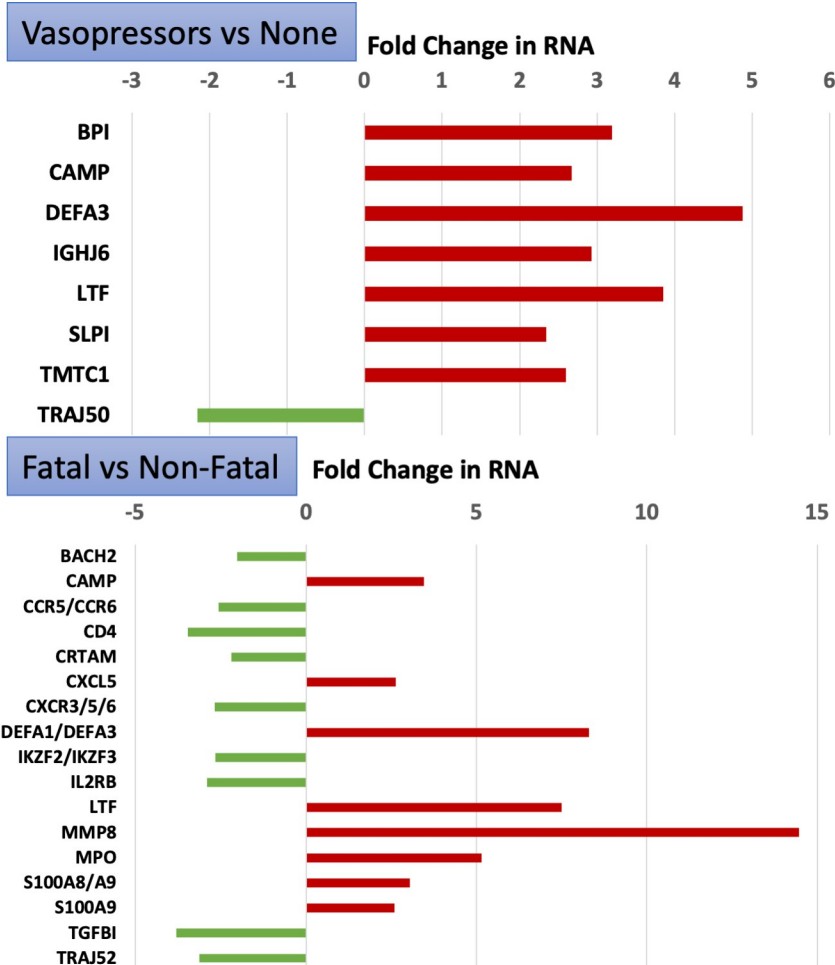

**Fig 2. RNAseq of COVID-19 patients stratified by vasopressor use or fatality.** The RNAseq data shown in Fig 1 was reanalyzed according to the severity of the COVID-19 using vasopressor use (upper panel) or fatality (lower panel) as the group criteria. DEGs were identified and examples of known transcripts are graphed with the fold-change increase (red bars) or decrease (green bars) of the patients relative to controls. Transcript gene symbols are shown on the left.

+ neutrophil RNA, and the buffy coat RNA were compared for levels of DEFA1, ALPL, IL8RB, MPO1, and RSTN (Fig 5). Surprisingly, the levels of these markers, especially DEFA1, was essentially uncorrelated between whole blood and CD15+ cells (Pearson R = 0.015, Fig 5, upper right panel). ALPL (R = 0.53) showed a more positive, but still modest correlation between the 2 sample types (upper left panel), while IL8RB (R = 0.04) was also uncorrelated between whole blood and purified CD15+ neutrophils (not shown). While the average absolute level of DEFA1 RNA was actually lower in CD15+ neutrophils than whole blood (20.4% vs. 68%), the ALPL (20.1% vs. 10.1%) and IL8RB levels (42.7% vs. 17.7%) were enriched in the CD15+ cells, consistent with the known localization of those markers to neutrophils.

Surprisingly, the buffy coat RNA demonstrated the best correlation to whole blood RNA (Fig 5, lower panels). Comparison of CD15+ neutrophil RNA to buffy coat RNA from the same subjects showed a relatively strong correlation between DEFA1 levels (r = 0.74), ALPL (0.51), and IL8RB (0.41), suggesting the RNA infection signals transcend the neutrophil subset. Buffy coat RNA exhibited a DEFA1 mRNA level about as strong as the whole blood of the same patients (58.9% vs 68% in whole blood). The ALPL (1.52%) and IL8RB (2.60%) signals

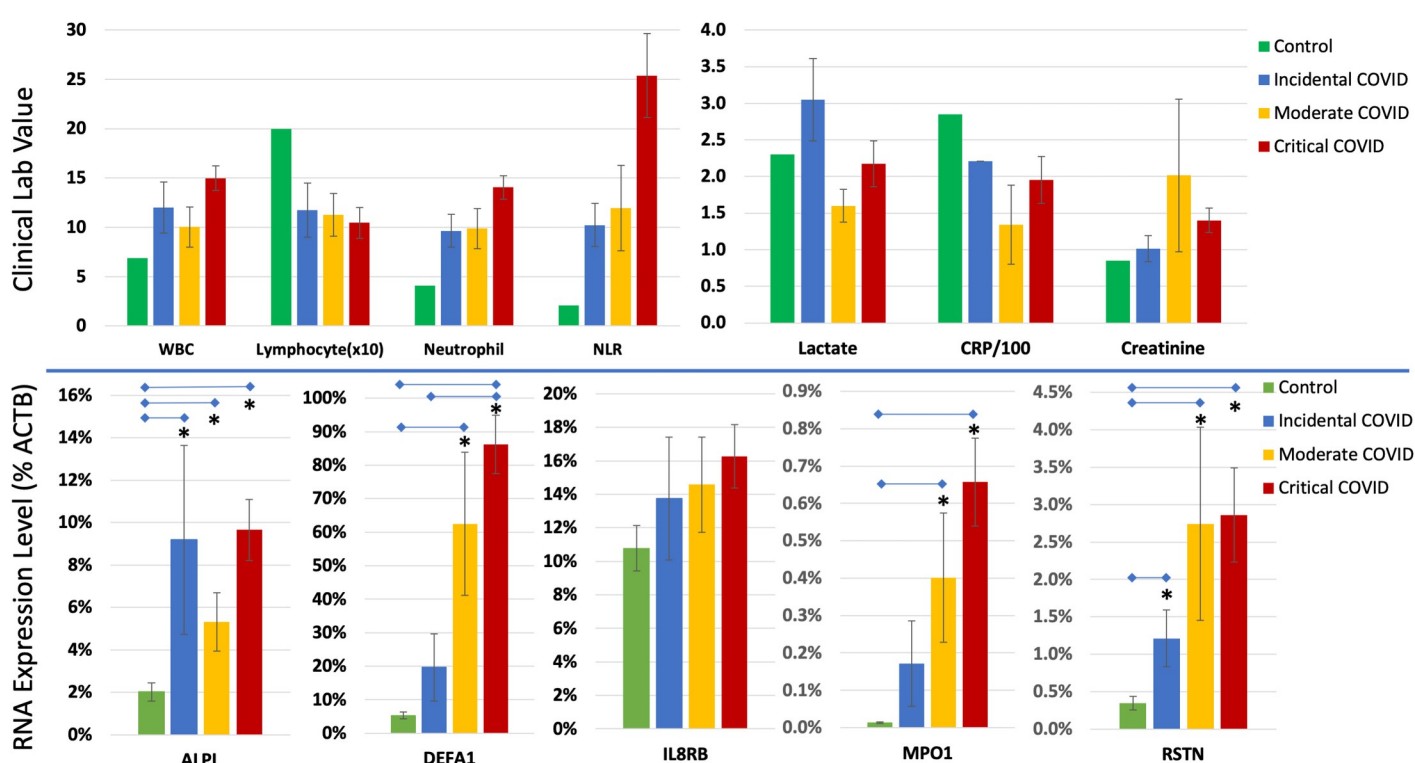

**Fig 3. Clinical and RNA measures of COVID-19 severity.** *Upper panels*: Patients testing positive for SARS-CoV2 by nasal swab PCR (n = 38) were divided into 3 groups of varying COVID-19 severity from Incidental (n = 7, asymptomatic and not the primary reason for admission), Moderate (n = 7, symptomatic but not on vasopressors or intubated), and Critical (n = 24, either vasopressors, intubated, or fatal). Reference levels (Control) are derived from published studies of normal controls. Bars reflect the white blood count (WBC), lymphocyte count (x10), neutrophil count, neutrophil/lymphocyte ratio (NLR), blood lactate, C-reactive protein (CRP), and creatinine for each group (mean ± s.e.m.). *Lower panels*: The same groups were used to calculate blood RNA levels for 5 transcripts, alkaline phosphatase (ALPL), defensin A1 (DEFA1), interleukin 8 receptor beta (IL8RB), myeloperoxidase 1 (MPO1), and resistin (RSTN), as a percentage of the actin B1 (ACTB) transcript used as the reference (i.e., ALPL in Incidental COVIDS is 9% of ACTB1 levels in copies per 20 μl of RNA in blood by ddPCR). Control subjects are unaffected normal subjects (n = 7) tested by the same process. (* = p < 0.05).

were strikingly lower in buffy coat compared to the CD15+ neutrophils (ALPL = 20.1%, IL8RB = 42.7%) or whole blood RNA (ALPL = 10.1%, IL8RB = 17.7%), consistent with the belief that they are relatively 'neutrophil-specific' markers.

There are several potential explanations for the strong correlation of the buffy coat RNA levels compared to neutrophil levels of these RNA biomarkers. First, some neutrophils, especially immature 'band' neutrophils, segregate into the buffy coat, as opposed to the heavier neutrophils that sediment with RBC [24]. Secondly, the buffy coat contains other cells such as monocytes which could express these RNAs, despite being considered 'neutrophil-specific' transcripts. Third, the CD15+ cells undergo a relatively lengthy isolation process of about 1 hour in which they are viable and could be inducing or degrading particular RNAs. By comparison, the buffy coat isolation is relatively fast, and might retain the whole blood signal better than purified PMNs.

## CD15+ cells from COVID-19 patients show increased nuclear chromatin staining

Based on the elevated neutrophil RNA levels of DEFA1, it was hypothesized that CD15+ neutrophils from COVID-19 patients might show visible signs of cellular activation. Affinity purified CD15+ neutrophils were isolated from a subset of patients with or without COVID-19

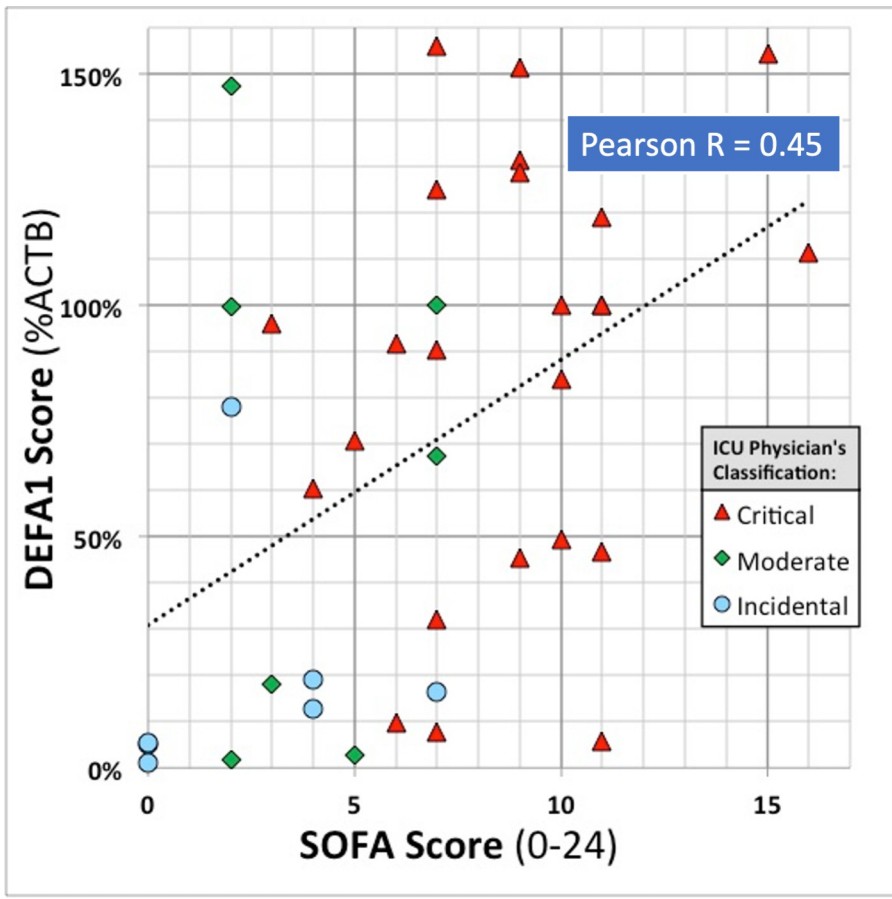

**Fig 4. Whole blood DEFA1 RNA vs SOFA score of COVID-19 severity.** Blood RNA from COVID-19 patients in the ICU (n = 38) was quantified for DEFA1 mRNA level by ddPCR and expressed as a percent of the ACTB mRNA level in the fixed RNA sample amount per patient (200 ng) thus minimizing changes in the abundance of neutrophils in blood between patients. The DEFA1 level (Y axis, as %ACTB) is plotted against the SOFA score determined clinically in the ICU according to the number and of degree of organ distress in the patient. Each point reflects a patient, colored according to their clinical classification. Dashed line indicates a linear fit to the data. R = Pearson's correlation between SOFA and DEFA1.

and then magnetically adhered to glass slides for histological analysis. Vital (Hoechst 33342) and fixed staining (DAPI) of the nuclear chromatin were performed on CD15+ neutrophils from COVID-19 patients. Patients from the groups with more severe symptoms showed increased neutrophil numbers and increased intensity of nuclear chromatin staining, suggesting increased accessibility of the chromatin to uptake the stain (Fig 6). To our knowledge, this is the first report of heightened nuclear fluorescence (DAPI^bright) staining of neutrophils in relation to human infection.

## CD15+ cells from COVID-19 patients demonstrate elevated neutrophil elastase activity

Nuclear chromatin decompaction is partially attributed to translocation of neutrophil elastase from granules to the nucleus, and is one of the early steps in the assembly and release of neutrophil extracellular traps (NETs). Elastolytic enzymes, such as neutrophil elastase (NE) and cathepsin G, cleave the histones that maintain chromatin packing. Prior studies have shown that biomarkers of NETosis, such as cell free DNA (cfDNA), extracellular histone H3, and NE

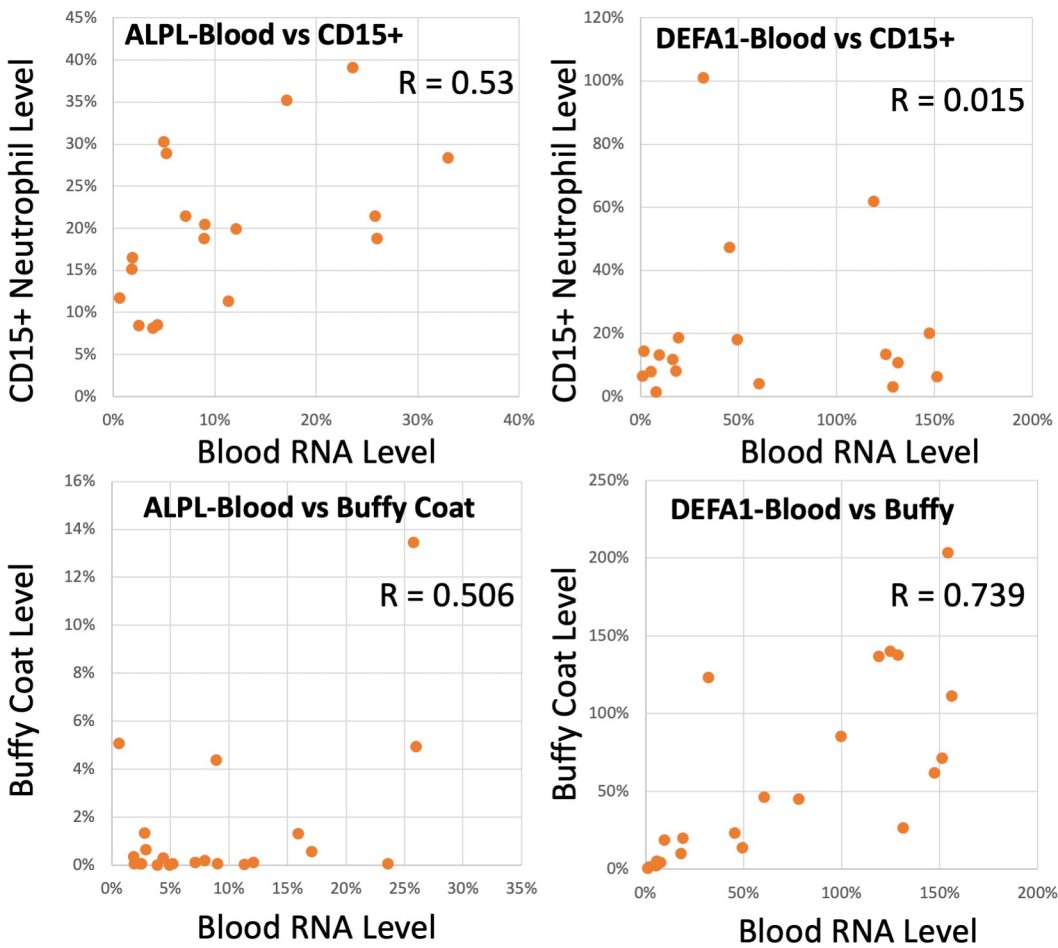

**Fig 5. Whole blood RNA levels vs purified CD15+ neutrophil or buffy coat levels of immune activation markers.** In a subset of COVID-19 patients, whole blood RNA was compared to RNA from purified CD15+ neutrophils (n = 16) or buffy coat (n = 22) from the same patients for 6 mRNA biomarkers using ddPCR quantitation. Each point represents a single patient quantified for ALPL (left panels) or DEFA1 (right panels) RNA in whole blood RNA vs purified CD15+ neutrophils (upper panels) or buffy coat RNA (lower panels). All RNA levels are expressed as % of ACTB in the same ddPCR measurement.

are elevated in COVID19 patients and correlate with the severity of disease [25]. Thus, the hypothesis that NE activity was increased by COVID-19 infection was tested *ex vivo*. NE activity was measured in purified, lysed CD15+ neutrophils using a kinetic assay that measured the cleavage of a fluorescent peptide substrate over time (Fig 7). The number of CD15+ neutrophils assayed was equivalent to the number of neutrophils in a fingerstick volume (50 μl) of the patient's whole blood. Not surprisingly, the number of CD15+ cells per 50 μl was 5.2-fold higher in COVID-19 patients (n = 10, avg. = 7.05 x $10^5$ cells) than in in controls (n = 4, avg. = 1.35 x $10^5$ cells; p = 0.25). The elastase activity in the COVID-19 patients (4 shown) was markedly higher than 2 uninfected control subjects (Fig 7, left panel). In parallel to the NE assay, the same COVID patients' CD15+ cells were chromatin-stained at 30 minutes, and they demonstrated striking adhesion, spreading, and NETosis, as shown for one patient (right panel).

As shown in Fig 8, total elastase activity in purified CD15+ cells was increased more than 13-fold in COVID-19 patients (n = 10) relative to Controls (n = 4, p = 0.008). Compensating for the increased numbers of neutrophils in COVID-19 patients, which was elevated about

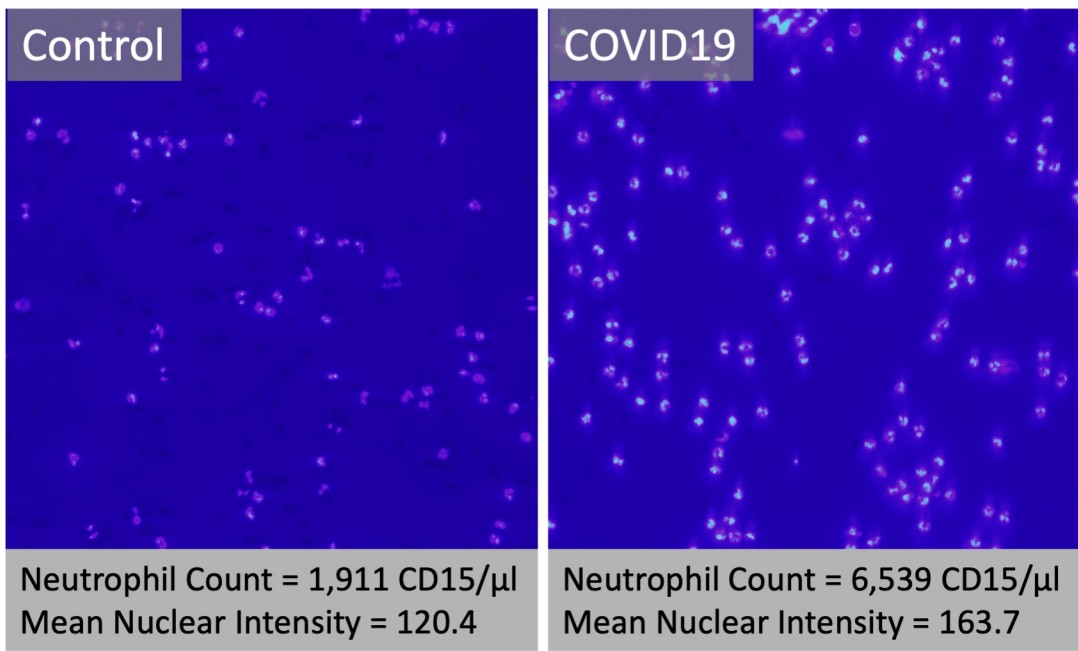

**Fig 6. Fluorescent DNA stain of purified CD15+ neutrophils from control or COVID-19 patients.** Purified CD15 + neutrophils were purified from whole blood of Control (left panel) or COVID-19 patients (right panel) using antibody-coated paramagnetic beads. Washed cells were bound magnetically to a glass slide, briefly fixed, and stained with DAPI, which fluoresces only when intercalated on double strand DNA. The neutrophil count and mean nuclear intensity were quantified for digital images at identical exposures using image analysis (ImageJ, NIH). Reported values reflect mean of 3 random 20X fields per subject.

3-fold in this larger group (Fig 8A), demonstrates that CD15+ neutrophils had a 4.7-fold increased neutrophil elastase activity per cell ($p = 0.012$). This is a surprising and potentially important finding because it was not previously known that the elastase activity per cell increased in human viral infections. Thus, the increased elastase observed in COVID-19 patients is due to both an increase in the number of CD15+ neutrophils, and an increase in the elastase activity produced by each cell.

The level of elastase activity in the purified CD15+ neutrophils appeared to be related to the severity of the infection, as shown in Fig 8B. Comparing COVID-19 patients that did not need vasopressor support ($n = 8$) with those that did need vasopressor support ($n = 4$) indicates that while the CD15+ cell count was not significantly different between groups (355K vs 372K, respectively), the neutrophil elastase activity per cell was increased >2-fold in the patients needing vasopressor support ($p < 0.05$). By comparison, in this same smaller group of patients, neither the NLR (no pressor = 10.82 vs pressor = 6.57, $n = 7$ vs 3, ns) nor the lactate levels (no pressor = 2.25 vs pressor = 2.13, $n = 4$ vs 3, ns) were different between no pressors and pressor patients.

The clinical consequence of increased elastase activity in COVID-19 patients is not clear. While enhanced NE activity might help to clear extracellular SARS-CoV2, other studies indicate that human neutrophil elastase's cleavage of the spike protein within an activation loop region may enhance fusion of the viral envelope with host cell membranes [26]. NE levels in plasma, and other NETosis markers, are elevated in COVID-19 cases and may have prognostic value toward severity [3, 11, 25]. However, NE activity may be more than a biomarker of neutrophil activation because early mutations in the SARS-CoV2 spike protein created elastase cleavage sites that seem to enhance viral spread [27, 28]. Overall, functional analysis suggests that the virus does not replicate well in neutrophils [29].

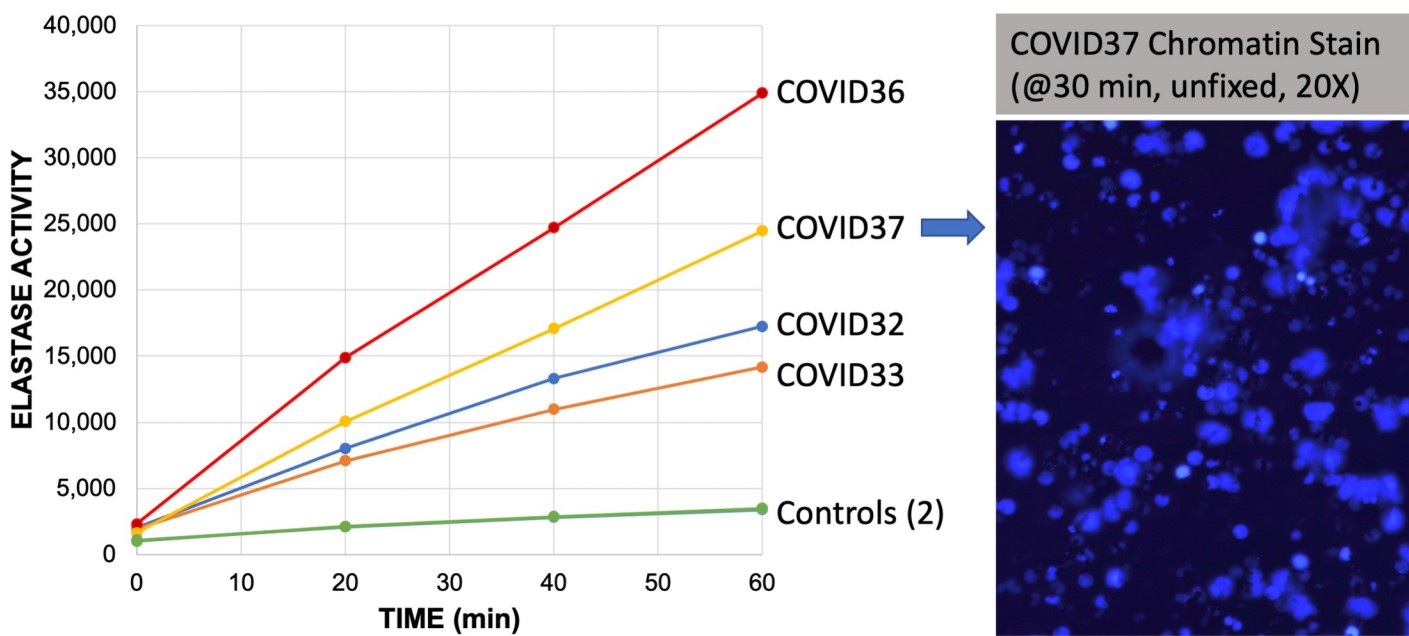

**Fig 7. Neutrophil elastase activity of purified CD15+ neutrophils from control or COVID-19 patients.** CD15+ neutrophils were purified from whole blood of patients or controls using antibody coated magnetic beads, which were then counted using a hemocytometer. The CD15+ neutrophils from the equivalent of 50 μl of blood (250K-600K cells) were then freeze-thawed (5X) and assayed for elastase activity using a fluorometric substrate measured over time for 4 COVID-19 patients (i.e. COVID36, COVID37, etc.). Each point is a single measure of fluorescence at the specified times after adding substrate. Controls (n = 2) are essentially superimposed. In the right panel, parallel samples of CD15+ neutrophils were magnetically adhered to a glass slide in HBSS and allowed to attach for 30 min prior to staining with DAPI to image the DNA without fixation. Photomicrograph is stimulated at 375 nm and emission captured at 415–460 nm with a 20X objective.

## Plasma elastase activity in COVID-19 patients

Because the CD15+ neutrophils had elevated elastase activity in COVID-19 patients, it raised the possibility that circulating plasma levels of elastase might also be elevated, making a convenient biomarker of COVID-19 infection. Neutrophil elastase protein, measured antigenically in plasma, was increased in COVID-19 patients [11, 29]. However, in a subset of the current cohort, the elastase activity in their plasma did not correlate well with their status as control or COVID-19 patients (Fig 9). The plasma (50 μl) was used in equivalent volume to the CD15 + cells, and was treated identically with respect to lysis method (freeze-thaw), substrate concentration, and time. Overall, from analysis of 5 COVID patients, with known high elastase activity in CD15+ cells, versus 3 control patients, elastase activity in plasma was detected at only 1% of the level detected in the plasma (Fig 9).

## Limitations of the current study

Because of the difficulty in consenting and studying severely ill COVID-19 patients in the midst of the pandemic, the sample size for the current study is modest. As our studies began to identify NE as a potential biomarker and point-of-care assay, the vaccines became available and were prevalent in the DC area, fortunately reducing the number of COVID-19 cases seen at our hospital. A surprising limitation was the variable certainty with which the clinical syndrome could be attributed to SARS-CoV2 infection. Some patients that were asymptomatic and were only included because of positive SARS-CoV2 nasal swab as a part of routine surveillance, turned out to have very high blood titer of virus. Conversely, some very clinically ill patients did not have high viral RNAemia, which could indicate that the virus was highly

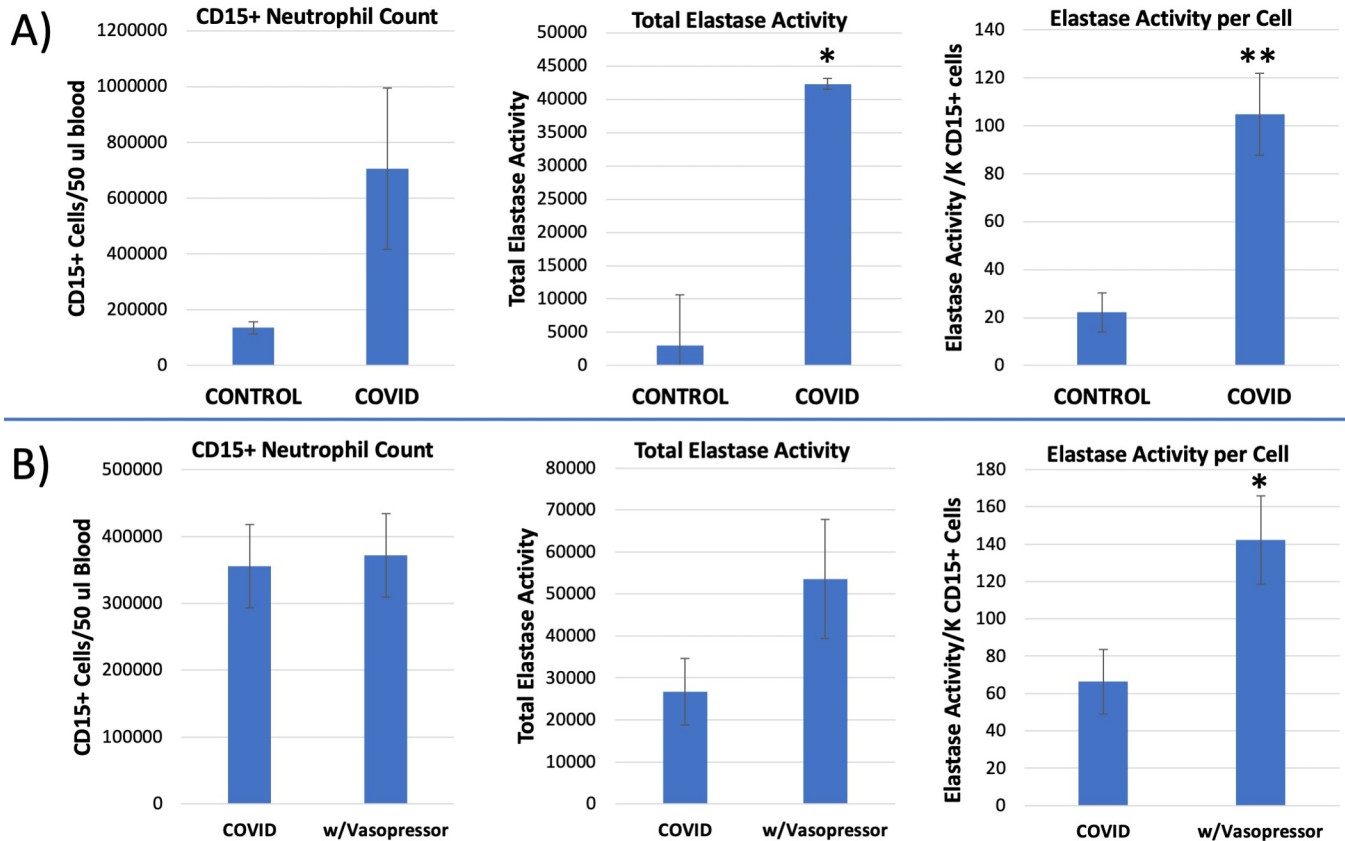

**Fig 8. Neutrophil elastase activity of purified CD15+ neutrophils from control or COVID-19 patients. Panel A)** Using the kinetic elastase assay shown in Fig 7, COVID-19 patient's purified CD15+ neutrophils (n = 10) were compared to control subjects (n = 4). The left panel reports the total cell count of CD15 + neutrophils from 50 μl of blood (mean ± s.e.m.). The total elastase activity of those cells was measured in arbitrary fluorescence units per 2 hour incubation (middle panel, mean ± s.e.m.). The total elastase activity per patient was divided by the CD15+ cell count for that patient to yield the elastase activity per cell (right panel, mean ± s.e.m.). **Panel B)** Patients that were PCR+ for SARS-CoV2, were regrouped according to whether they received vasopressor support (n = 4) or did not receive vasopressors (n = 8), (* = p<0.05, ** = p<0.01).

localized in the lungs, for instance, or that the RNAemia had cleared, but secondary, opportunistic infections had prevailed.

## Conclusions

COVID-19 syndrome triggers potent changes in the blood transcriptome as measured by RNAseq, and confirmed on selected targets by ddPCR. The severity of COVID-19 is associated with significant changes in the activation of neutrophil-related transcripts, such as DEFA1, and decreased levels of T cell-related transcripts, such as the T cell receptor. Seropositivity for SARS-CoV2, as measured by ddPCR of blood RNA, was detected at 95.5% sensitivity by an elevated DEFA1 RNA in blood. Purification of CD15+ neutrophils from COVID-19 patients did not markedly enrich for these 'neutrophil-specific' transcripts, suggesting other cell types may contribute to their elevation in the whole blood RNA of infected persons. Isolated CD15+ neutrophils from COVID-19 patients showed striking increases in nuclear DNA staining with DAPI, consistent with the early steps of chromatin remodeling in NETosis. The COVID-19 neutrophils likewise exhibited >5-fold elevated neutrophil elastase activity on a per-cell basis, consistent with the elevated transcript levels of cathepsin G (CTSG) and DEFA1, and the decompacted, DAPI^bright chromatin. Circulating plasma levels of elastase activity in COVID-19

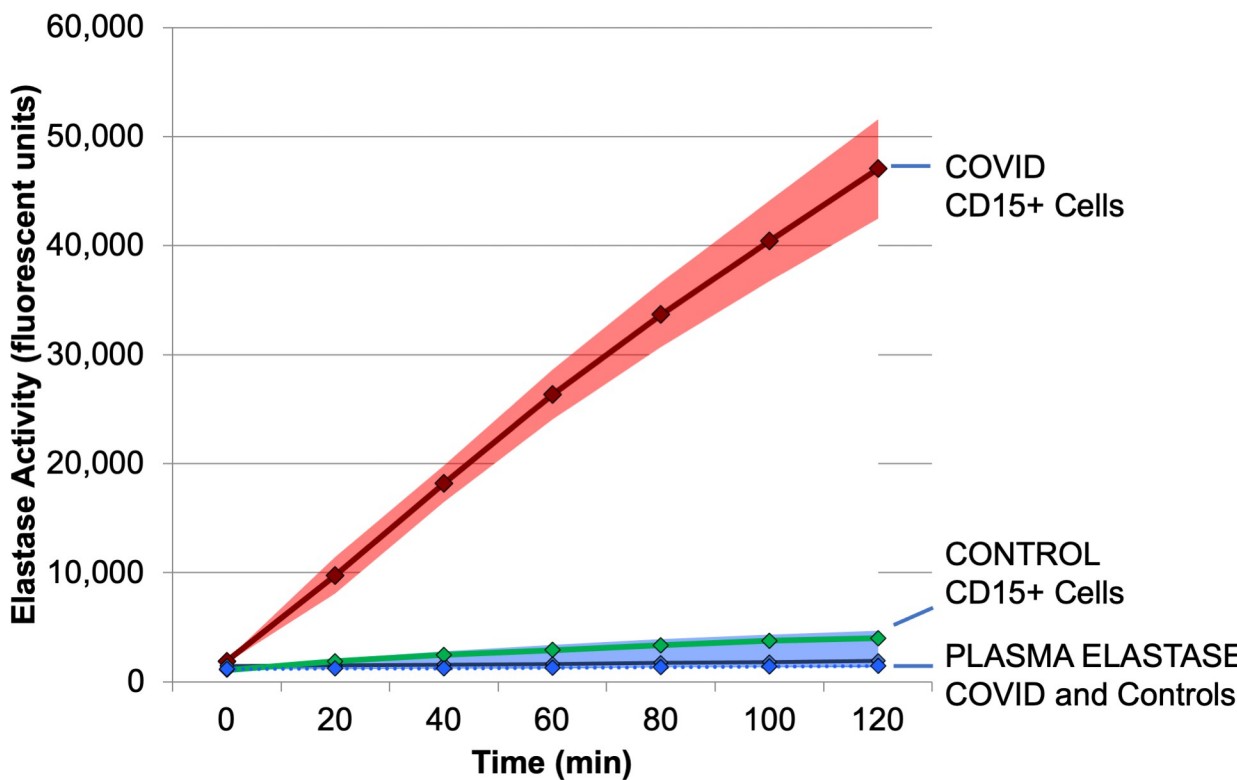

**Fig 9. Neutrophil elastase activity of purified CD15+ neutrophils vs. autologous plasma from control or COVID-19 patients.** Elastase activity, in relative fluorescence units (Y Axis) was measured in freeze-thaw lysates of purified CD15+ neutrophils from COVID-19 patients (n = 5, RED line) versus plasma (n = 5, BLUE solid line) from the same patients. Points reflect the mean fluorescence (Y axis) at each time point (X axis) with 95% confidence intervals in light red or blue shading. For reference, elastase active from CD15+ neutrophils (GREEN) or plasma (BLUE dashed) of normal controls (n = 3). Plasma elastase from COVID-19 and Controls are low and nearly superimposed.

patients was very low (~1%) compared to CD15+ neutrophil elastase activity of the same patients. Neutrophil elastase has the potential to be useful as a rapid point-of-care assay for the detection of neutrophil activation.

## Supporting information

**S1 Table. Additional demographic parameters of subjects with statistical analysis.**
(XLSX)

**S2 Table. All differentially expressed genes (DEGs) between COVID and Controls.**
(XLSX)

**S3 Table. Complete RNAemia DEGs summarized in Table 2.**
(XLSX)

**S4 Table. DEGs related to vasopressor status.**
(XLSX)

**S5 Table. DEGs related to fatality from COVID19.**
(XLSX)

**S6 Table. DEGs matching prior report by Notaborollo et al.**
(XLSX)

## Acknowledgments

The authors are deeply grateful to the COVID-19 patients and their families who kindly agreed to participate in this research study despite life-changing complications. Our most sincere respect and gratitude goes out to the GW ICU nursing staff and physician assistants, and all the healthcare professionals involved in diagnosing and treating the COVID-19 patients at GW and around the world.

## Author Contributions

**Conceptualization:** John LaFleur, David Yamane, Ivy Benjenk, Eric Heinz, Ian Toma, Timothy A. McCaffrey.

**Data curation:** Richard Wargodsky, Philip Dela Cruz, John LaFleur, David Yamane, Justin Sungmin Kim, Ivy Benjenk, Eric Heinz, Obinna Ome Irondi, Katherine Farrar, Ian Toma, Tristan Jordan, Jennifer Goldman, Timothy A. McCaffrey.

**Formal analysis:** Richard Wargodsky, Philip Dela Cruz, John LaFleur, David Yamane, Justin Sungmin Kim, Obinna Ome Irondi, Ian Toma, Tristan Jordan, Jennifer Goldman, Timothy A. McCaffrey.

**Funding acquisition:** Timothy A. McCaffrey.

**Investigation:** Richard Wargodsky, Philip Dela Cruz, John LaFleur, David Yamane, Justin Sungmin Kim, Ivy Benjenk, Eric Heinz, Obinna Ome Irondi, Katherine Farrar, Ian Toma, Tristan Jordan, Jennifer Goldman, Timothy A. McCaffrey.

**Methodology:** Richard Wargodsky, Philip Dela Cruz, John LaFleur, David Yamane, Ivy Benjenk, Eric Heinz, Katherine Farrar, Ian Toma, Tristan Jordan, Jennifer Goldman, Timothy A. McCaffrey.

**Project administration:** Richard Wargodsky, John LaFleur, David Yamane, Justin Sungmin Kim, Ivy Benjenk, Katherine Farrar, Ian Toma, Timothy A. McCaffrey.

**Resources:** David Yamane, Timothy A. McCaffrey.

**Supervision:** Philip Dela Cruz, John LaFleur, David Yamane, Justin Sungmin Kim, Eric Heinz, Ian Toma, Timothy A. McCaffrey.

**Validation:** Philip Dela Cruz, David Yamane, Ian Toma, Timothy A. McCaffrey.

**Visualization:** John LaFleur, Obinna Ome Irondi, Tristan Jordan, Timothy A. McCaffrey.

**Writing – original draft:** John LaFleur, Timothy A. McCaffrey.

**Writing – review & editing:** Richard Wargodsky, Philip Dela Cruz, John LaFleur, David Yamane, Justin Sungmin Kim, Ivy Benjenk, Eric Heinz, Obinna Ome Irondi, Katherine Farrar, Ian Toma, Tristan Jordan, Jennifer Goldman, Timothy A. McCaffrey.

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
