## [Decision Letter · Decision Letter 0]

18 Nov 2021

PONE-D-21-30161RNA Sequencing in COVID-19 patients identifies neutrophil activation biomarkers as promising diagnostic platform for infections.PLOS ONE

Dear Dr. Mc Caffery.

Thank you for submitting your manuscript to PLOS ONE. After careful consideration, we feel that it has merit but does not fully meet PLOS ONE’s publication criteria as it currently stands. Therefore, we invite you to submit a revised version of the manuscript that addresses the points raised during the review process. The reviewer has raised major concerns on the data presentation and analysis.Kindly make changes as requested.

 Please submit your revised manuscript by Jan 02 2022 11:59PM. If you will need more time than this to complete your revisions, please reply to this message or contact the journal office at plosone@plos.org. Please include the following items when submitting your revised manuscript:A rebuttal letter that responds to each point raised by the academic editor and reviewer(s). You should upload this letter as a separate file labeled 'Response to Reviewers'.A marked-up copy of your manuscript that highlights changes made to the original version. You should upload this as a separate file labeled 'Revised Manuscript with Track Changes'.An unmarked version of your revised paper without tracked changes. You should upload this as a separate file labeled 'Manuscript'.

We look forward to receiving your revised manuscript.

Kind regards,

Afsheen Raza, PhD

Academic Editor

PLOS ONE

2. Please provide additional details regarding participant consent. In the ethics statement in the Methods and online submission information, please ensure that you have specified what type you obtained (for instance, written or verbal, and if verbal, how it was documented and witnessed). If your study included minors, state whether you obtained consent from parents or guardians.

“The authors are grateful for the financial support of True Bearing Diagnostics, Inc. and

The St. Laurent Institute. The authors are also grateful for the institutional support provided by

the CTSI-CN Award Number UL1TR001876 from the NIH National Center for Advancing

Translational Sciences, and Core Instrument Grant for the Bio-Rad ddPCR S10 OD021622.”

We note that you have provided funding information within the Funding Section. Please note that funding information should not appear in other areas of your manuscript. We will only publish funding information present in the Funding Statement section of the online submission form.

“The authors are grateful for the financial support of True Bearing Diagnostics, Inc. and

The St. Laurent Institute. The authors are also grateful for the institutional support provided by

the CTSI-CN Award Number UL1TR001876 from the NIH National Center for Advancing

Translational Sciences, and Core Instrument Grant for the Bio-Rad ddPCR S10 OD021622.”

“The authors are grateful for the financial support of True Bearing Diagnostics, Inc. and

The St. Laurent Institute. The authors are also grateful for the institutional support provided by

the CTSI-CN Award Number UL1TR001876 from the NIH National Center for Advancing

Translational Sciences, and Core Instrument Grant for the Bio-Rad ddPCR S10 OD021622.”

“TM and IT have an equity interest in True Bearing Diagnostics, Inc., a diagnostics company developing RNA biomarkers for various diseases, including coronary artery disease and internal infections.  TM is seeking patent protection for technology related to the current studies.  The other authors declare there are no competing interests.”

6. Please include captions for your Supporting Information files at the end of your manuscript, and update any in-text citations to match accordingly. Please see our Supporting Information guidelines for more information: http://journals.plos.org/plosone/s/supporting-information

Reviewers' comments:

Reviewer's Responses to Questions

**Comments to the Author**

1. Is the manuscript technically sound, and do the data support the conclusions?

Reviewer #1: Yes

2. Has the statistical analysis been performed appropriately and rigorously? 

Reviewer #1: N/A

3. Have the authors made all data underlying the findings in their manuscript fully available?

Reviewer #1: Yes

4. Is the manuscript presented in an intelligible fashion and written in standard English?

Reviewer #1: Yes

5. Review Comments to the Author

Reviewer #1: The authors set out to identify biomarkers associated with Covid-19 disease severity. They measured the host response to the infection via blood transcriptome profiling. Their analysis rapidly focused on sets of transcripts associated with neutrophil activation. From there they carried out further work profiling the transcriptome of neutrophils isolated from Covid-19 patients and controls. Downstream assays were also run including DNA staining and the measurement of neutrophil elastase activity, as well as measurement elastase activity in serum. The authors conclude that abundance levels in whole blood of DEFA1 transcripts and neutrophil elastase activity may serve as clinically relevant markers in the context of the management of patients with Covid-19 in particular.

While the study could make a significant contribution to Covid-19 research several limitations should be addressed:

Major

- The work is not well-grounded in the literature. Running the query “Covid-19 AND Netosis” returns 69 articles, which include several which are of direct relevance and are not currently cited [e.g. 34344929]. “Covid-19 AND Neutrophils AND Elastase” returns 36 articles. Some important work investigating the role of neutrophils in viral respiratory infection and association with severity is not discussed either [e.g. 29777224].

- The number of subjects used in comparisons is often very small, which inevitably undermines some of the conclusions. There is probably not much to be done at this stage (unless available public provided opportunities for re-use / independent validation). But it would be preferrable not to subdivide already small study groups (e.g in some instances, n=2, n=4, n=5, n=7 etc…). Just for context, this study, which is not yet publish but seeks to address similar questions comprises > 300 subjects. Given the extent of inter-individual variability in patient cohorts it is the kind of numbers that would permit to bring more convincing answers. In any case this limitation should at least be discussed.

- The transcriptome analysis was very cursory. It seems the authors had decided from the onset to focus on neutrophils markers. The paper would benefit from a more global analysis being performed before deep diving on neutrophil related genes and neutrophils.

- The quality of the figures is generally not very good, and they are not always informative. The DAPI staining images are not very clear and the labels that are overlaid obstruct a good part of the image. The use of an arrow on the line graph on Figure 9 is somewhat unusual for an original research paper. There are currently 9 figures, and the paper would likely benefit from focusing on those that are considered more critical to the message being conveyed.

- Before acceptance should be deposited in GEO and token provided for reviewer access (main point here would be to verify that the level of details provided is sufficient)

6. PLOS authors have the option to publish the peer review history of their article (what does this mean?). If published, this will include your full peer review and any attached files.

Reviewer #1: No

---

## [Editor Report · Decision Letter 1]

9 Dec 2021

RNA Sequencing in COVID-19 patients identifies neutrophil activation biomarkers as promising diagnostic platform for infections.

PONE-D-21-30161R1

Dear Dr. McCaffrey,

We’re pleased to inform you that your manuscript has been judged scientifically suitable for publication and will be formally accepted for publication once it meets all outstanding technical requirements.

Kind regards,

Afsheen Raza, PhD

Academic Editor

PLOS ONE
---

## [Editor Report · Acceptance letter]

31 Dec 2021

PONE-D-21-30161R1 

RNA Sequencing in COVID-19 patients identifies neutrophil activation biomarkers as a promising diagnostic platform for infections. 

Dear Dr. McCaffrey:

I'm pleased to inform you that your manuscript has been deemed suitable for publication in PLOS ONE. Congratulations! Your manuscript is now with our production department. 

Kind regards, 

on behalf of

Dr. Afsheen Raza 

Academic Editor

PLOS ONE